# A Novel Field-Free Line Generator for Mechanically Scanned Magnetic Particle Imaging

**DOI:** 10.3390/s24030933

**Published:** 2024-01-31

**Authors:** Tae Yi Kim, Jae Chan Jeong, Beom Su Seo, Hans Joachim Krause, Hyo Bong Hong

**Affiliations:** 1Electronics and Telecommunications Research Institute (ETRI), 218 Gajeong-Ro, Yuseong-Gu, Daejeon 34129, Republic of Korea; jinsun@etri.re.kr (T.Y.K.); channij80@etri.re.kr (J.C.J.); bsseo@etri.re.kr (B.S.S.); 2Institute of Biological Information Processing, Bioelectronics (IBI-3), Forschungszentrum Jülich, 52425 Jülich, Germany

**Keywords:** field-free line, magnetic particle imaging, frequency mixing magnetic detection

## Abstract

In this study, we propose an efficient field-free line (FFL) generator for mechanically driven FFL magnetic particle imaging (MPI) applications. The novel FFL generator comprises pairs of Halbach arrays and bar magnets. The proposed design generates high-gradient FFLs with low-mass permanent magnets, realizing fine spatial resolutions in MPI. We investigate the magnetic field generated using simulations and experiments. Our results show that the FFL generator yields a high gradient of 4.76 T/m at a cylindrical field of view of 30 mm diameter and a 70 mm open bore. A spatial resolution of less than 3.5 mm was obtained in the mechanically driven FFL-MPI.

## 1. Introduction

Magnetic particle imaging (MPI) is a tracer imaging technique that detects the positions and densities of injected magnetic particles via the nonlinear magnetic response [1]. Iron-oxide tracers are safe for humans and circulate through the bloodstream without producing any ionizing radiation. Thus, MPI has attracted significant research attention with regard to medical applications [2,3].

Existing MPI techniques usually require a fine spatial resolution, which largely depends on the gradient strength of the selection field [4]. The selection field is an inhomogeneous magnetic field with an unsaturated magnetic field area, wherein the magnetization change caused by a drive field is generated selectively [5]. A field-free point (FFP) or a field-free line (FFL) are usually employed for selection fields. As the gradient of the selection field increases, the size of the region in which the tracers respond to the excitation field decreases [6]. This results in a narrower spatial resolution in the MPI.

FFL systems have been developed using current-driven coils, permanent magnets, or combinations of the two [2,3,6,7,8]. Coil-based FFL generators are advantageous in that it is easy to construct open bores, and fast scanning can be achieved with rapid control of the magnetic field strength via a current. However, coil-based FFL generators consume a large amount of power when shifting the high-gradient selection field, even for small bore sizes [9]. Recently, Halbach arrays have been utilized in FFL generators for MPI. Halbach arrays use permanent magnets instead of electromagnets, which reduces the power consumption [10,11,12]. However, the FFL is moved electronically by current-driven coil configurations, involving high power consumption when the field of view (FOV) is larger. The high power consumption is a critical issue to be resolved in developing FOVs suitable for imaging large sections of the human body. Additionally, the large amplitude and high frequency required for the drive field raise severe health concerns and generate significant noise [13].

The scanning speed of mechanically driven MPI is lower than that of electronically controlled MPI. Nevertheless, recent high-tech motor stages are sufficiently fast and robust that 2D MPI scanning can be performed in a few seconds. Despite the time required to achieve a high temporal resolution, MPI is still useful in diverse applications. For example, recent studies have investigated MPI as a continuous clinical monitoring tool for substituting conventional CT and MRI [14,15]. Notably, CT cannot be employed for recurrent monitoring owing to radiation exposure, and MRI entails a substantially long examination time [16,17]. Consequently, the imaging time of MPI, which is only a few minutes, will not be a critical limitation in fields that require long-term, repeated monitoring.

Here, we propose a novel FFL generator for mechanically driven FFL-MPI based on pairs of Halbach arrays and bar magnets. In most MPI systems proposed to date, which use permanent magnets or a mixture of permanent magnets and electromagnets, the general method of increasing the gradient of the selection field has been to use a more powerful large magnet or to increase the number of magnets, such as using a Halbach magnet [8,14,18]. In this study, we were able to achieve a selection field of about 4.7 T/m by arranging relatively small-sized Halbach and bar magnets three-dimensionally. To the best of our knowledge, our permanent magnet-based FFL configuration is the first of its kind. Despite reducing the size and weight, the proposed FFL generator exhibits a high gradient with a large open bore.

This study also examines the magnetic field characteristics of the proposed FFL generator and the principle of the FFL formation and compares experimental data with simulation results. In addition, the linear and rotational movement characteristics of the FFL for mechanical FFL-MPI are explained. Finally, the MPI application results obtained in the mechanically driven manner based on this FFL generator are shown.

## 2. Materials and Methods

The proposed FFL generator is shown in Figure 1. It comprises pairs of Halbach ring-type arrays and bar-shaped magnets. For ease of construction, we set the bore size to 70 mm. The dimensions of the FFL device with a 70 mm open bore are detailed in Table 1, and we used a NdFeB (N35 grade) magnet. In this configuration, the field gradient is 4.76 T/m along the *x* and *z* axes. The detailed geometric values of this FFL generator are presented in Table 1.

To analyze the magnetic field of the proposed FFL generator and its movement, we performed simulations with the three-dimensional (3D) Faraday software (Faraday V10.2 Enginia Research Inc., Winnipeg, MB, Canada) using a computer with an Intel i7-8700K CPU and Windows 10 operating system. The FFL generator was constructed based on the simulation results, and the generated magnetic field was measured using a 3-axis Hall-effect sensor (TLX493D, Infineon, Neubiberg, Germany) connected to a robotic device (TT-C3-I-2020-10B, IAI Corporation, Shizuoka, Japan) translating a regular distance in three dimensions. A magnetic Hall sensor and a tabletop robot device are connected to a computer and controlled by lab-made software. The software controls the tabletop robot device’s movement along a desired path in desired intervals within a specified range. When the robot device arrives at a desired measurement location, the magnetic field is measured three times by the Hall sensor and the average value is stored along with the position.

Mechanically driven FFL-MPI was realized with rotational and translational stages (Namil Optical Instruments Co., Incheon, Republic of Korea). The excitation and detection of superparamagnetic iron-oxide nanoparticle (SPION) signals were evaluated based on the frequency mixing magnetic detection (FMMD) technique [6]. The experimental setup for MPI is shown in Figure 2. The Halbach array pair is mounted on the optical table, and the bar magnet pair is mounted on the rotational and translational (R&T) stages. The linear stage moves the sample into the FFL position inside the FMMD sensor. Signal generation and detection are carried out using an FMMD sensor reported previously [19,20]. The measured signal generates a sinogram, while the FFL scans the sample translationally and rotationally. Then, the sinogram data can be reconstructed into 2D images via the inverse Radon transform. The SPIONs used for the MPI sample were Synomag-D (product no. 104-00-501) from Micromod (Rostock, Germany). These nanoparticles have a diameter of 70 nm, a plain surface, and a concentration of 25 mg/mL. A thin capillary tube was filled with the SPIONs. The outer and inner diameters of the tube are 1.5 mm and 1.2 mm, respectively.

## 3. Results and Discussion

Figure 3a shows a photograph of the constructed FFL device. The magnets are positioned in an aluminum casing in accordance with the simulation dimensions listed in Table 1. The FFL generator and all casing components, such as bolts, are constructed from nonmagnetic materials, such as aluminum, so that the generated FFL magnetic field is not interfered with. The magnetic field was measured using a Hall-effect sensor positioned on a 3D moveable robotic arm. Figure 3b shows a comparison of the measured magnetic field along the *x*, *y*, and *z* axes with the simulation results. The scanning area was [–15 mm, 15 mm] along each axis at 5 mm intervals. The measured contour map is shown in Figure 3c. The measured data correspond closely with the simulation results.

The generation of the clear FFL of our new device, as shown in Figure 3, can be explained with the superposition principle of magnetic fields. The FFL generator is composed of the Halbach ring pair and the bar-shaped magnet pair, and the superposition of the magnetic fields of all the components explains the creation of an FFL.

The magnetic field simulation results for the Halbach ring pair and the bar-shaped magnet pair are shown in Figure 4d,f, respectively, in the *x*-*y*, *y*-*z*, and *x*-*z* planes near the center. The FFP field generated from the bar magnet and Halbach ring pairs can be approximated as
(1)$$H_{FFP}^{x}=\left(\begin{array}{ccc} G_{x_{h}} & 0 & 0\\ 0 & G_{y_{h}} & 0\\ 0 & 0 & G_{z_{h}} \end{array}\right)r_{0},\;\;H_{FFP}^{z}=\left(\begin{array}{ccc} G_{x_{b}} & 0 & 0\\ 0 & G_{y_{b}} & 0\\ 0 & 0 & G_{z_{b}} \end{array}\right)r_{0},$$
where $r_{0}={\left[x,y,z\right]}^{T}$ is the spatial position vector, *G* represents the gradient matrix, the subscripts *x*, *y*, and *z* denote the directions, and *h* and *b* denote the Halbach ring pair and the bar magnet pair, respectively.

The net magnetic field resulting from the superposition of the magnetic fields generated from the two components is expressed as
(2)$$H_{FFL}^{y}\left(r\right)=H_{FFP}^{x}\left(r\right)+H_{FFP}^{z}\left(r\right)=\left(\begin{array}{ccc} G_{x_{h}}+G_{x_{b}} & 0 & 0\\ 0 & G_{y_{h}+}G_{y_{b}} & 0\\ 0 & 0 & G_{z_{h}}+G_{z_{b}} \end{array}\right)r_{0}\;=\left(\begin{array}{ccc} G_{x_{h}}+G_{x_{b}} & 0 & 0\\ 0 & 0 & 0\\ 0 & 0 & G_{z_{h}}+G_{z_{b}} \end{array}\right)r_{0},$$
where $G_{y_{h}}\sim -G_{y_{b}}$.

When the $G_{y_{h}}$ has a similar magnetic field value to $G_{y_{b}}$ and the opposite vector direction, the superposition of them generates a zero field, as illustrated in Equation (2).

The newly proposed FFL generator is based on quadrupole magnets. Figure 5a shows that when the quadrupole magnets are stacked symmetrically to the *y* axis, the FFL is generated along the *y* axis. Figure 5e is a transformed design of Figure 5a where the square magnet pair of Figure 5a is substituted with a pair of Halbach rings. The magnetic field generated by the Halbach ring pair is similar to that of the square magnet pair, as shown in Figure 5b,f; therefore, an FFL is also generated by the magnet structure in Figure 5e.

We modified the cylindrical Halbach ring to a combination of small square magnets. This was done because both showed similar magnetic characteristic in a simulation study, but the use of square magnets provides more benefits in the context of an experiment. First, the cylinder-shaped magnets need to be custom-manufactured, whereas the square magnets are readily available on the market in a variety of sizes. Also, the radius is fixed when using cylinder-type magnets, but the combination of square magnets can easily be adjusted to the radius of the aperture by changing their number and their spacing. In addition, square magnets allow easier adjustment of magnet strength by overlapping several of them. Thus, square magnets were selected for Halbach ring pair fabrication instead of cylinder-type magnets. As a result, a clear straight FFL was found both in the simulation and the experiment, as shown in Figure 4 and Figure 5.

As a next step, we tried to check whether an MPI system that does not require a drive coil can be implemented using the above newly designed FFL generator. The important thing here is that the inner diameter of the Halbach ring pair is designed to the same value as the outer diameter of the FMMD sensor for MPI, so that the Halbach ring pair cannot move along the *x* and *y* axes for the movement of the FFL. Therefore, it is necessary to check whether the movement of the FFL is possible only through the movement of the bar magnet. In the simulation study with Faraday SW, the FFL movement and gradient strength were confirmed when just moving the bar magnet pair.

Figure 6a shows a schematic of the translational movement of the FFL. A simulation was conducted so that the bar magnet pair moves in the positive *x* direction up to 25 mm in increments of 5 mm. Figure 5b shows the two-dimensional simulation results of the magnetic field along the *x*-*y* plane. It can be seen that the FFL shifts to the traveling direction of the bar magnet while translating.

To determine whether the FFL field gradient is sustained when the bar magnet pair is displaced, the field gradient along the *x*, *y*, and *z* axes was investigated at each shift from 5 mm to 25 mm. Figure 6c,e,f show the field magnitudes along the *x*, *y*, and *z* axes. Figure 6d illustrates the overlapping data shown in Figure 6c, which were obtained to compare the field gradient along the *x* axis. As shown in Figure 6d–f, the gradient of the FFL was maintained with the same value when the bar magnet pair moved. Even though the *y* axis gradient shows a remnant field as the pair moves outside the center position, the value is much smaller than the saturation field strength of approximately 10 mT. Thus, the field gradient is negligible.

The displacement of the FFL center (*F_c_*) is linearly proportional to the bar magnet displacement (∆*x*), as shown in Figure 6c. The relation can be described by *y* = *ax*. This is because the gradient is the sum of the magnetic field due to the Halbach array pair and the bar magnet pair. The magnetic field of the bar magnet pair along the *x* axis near the zero position is approximately given by $y_{b}=G_{x_{b}}\;(x-∆x)$, and $y_{h}=G_{x_{h}}\;x$ for the Halbach array pair.

The sum of these components, which is the total magnetic field along the *x* axis, is given by
(3)$$y_{t}=G_{x_{b}}(x-∆x)+G_{x_{h}}x={(G}_{x_{b}}+G_{x_{h}})x-G_{x_{b}}∆x,$$

The relation between $F_{c}$ and ∆*x* is obtained by setting $y_{t}$ to zero and substituting *x* with $F_{c}$. The relation is given by
(4)$$F_{c}=G_{x_{b}}/{(G}_{x_{b}}+G_{x_{h}})∆x,$$

This explains the proportionality between *x* and $F_{c}$, and the slope is determined by $G_{x_{b}}$ and $G_{x_{h}}$. The gradients used in the simulation are $G_{x_{b}}\sim \;4$ and $G_{x_{h}}\sim \;1.25$, yielding *a* = 0.76. This result implies that the MPI raw data can be acquired with the same spatial resolution as that during the linear movement of the FFL.

The rotational movement of the FFL can be realized by rotating the bar magnet pair about the *z* axis. As shown in Figure 3b, the magnetic field of the Halbach ring pair along the *x-y* plane represents the isotropic features. This means that when the Halbach ring pair rotates along the *z* axis by any angle, the *x-y* plane field map shows the same results at any angle. Therefore, the rotation of the bar magnet can be considered as the rotation of the entire structure. As a result, the rotation angle of the bar magnet is the same as the rotation angle of the FFL.

The simulation results show that a sharp FFL can be generated via Halbach arrays and bar magnets, and the FFL can be linearly and rotationally moved simply by moving the bar magnet pair. These characteristics make the proposed FFL generator suitable for mechanically driven MPI applications. Above all, as the Halbach array can remain fixed during the MPI scanning, a compact and high-resolution MPI system can be developed. If the Halbach array has to move with both linear and rotational motions for MPI scanning, its aperture size should include additional space for linear motion without overlapping samples. In contrast, the FFL can be moved only via the movement of the bar magnet while the Halbach array is fixed; therefore, the aperture size of the Halbach array needs to be greater than the maximum diameter of the sample. When designing an FFL-generating device using a permanent magnet, it is advantageous to create a strong FFL gradient when the magnet is close to the FFL position. Consequently, it is beneficial to develop an FFL device with a stronger magnetic gradient and a high-resolution MPI system by using the combination of fewer magnets with a fixed Halbach array and a moving bar magnet.

Finally, we experimentally verified whether MPI images can be successfully obtained with the combination of the fixed Halbach array and the moving bar magnet, as shown by the above simulations. The spatial resolution of the MPI image was also checked. The configuration of the experimental setup is shown in Figure 2. The Halbach array was fixed to the optical table, and the pair of bar magnets were fixed to the R&T stage and moved linearly and rotationally with the stage. The speed of the translational and rotational movement of the R&T stage was 8.57 mm/s and 7.14 degrees/s, respectively.

The SPIONs were inserted into a very thin capillary tube, with the outer and inner diameters of 1.5 and 1.2 mm, respectively. To measure the resolution, MPI experiments were performed when the intervals of the capillary filled tube samples were 2 mm and 1 mm. As shown in Figure 7, when the distance between the samples was 2 mm, the two spaces were clearly separated in the sinogram. The sinogram was converted into a 2D image through inverse Radon transformation, and it was confirmed that the response of the SPIONs was clearly separated into two spots. When the distance between the samples was 1 mm, the response signals were only faintly separated in the sinogram; however, the boundary of the SPIONs was not clearly visible in the 2D transformed image, as shown in Figure 7b.

Thus, it has been shown for the first time that MPI images can be successfully obtained by combining a fixed Halbach array and a moving bar magnet. In this experiment, a new type of FFL generator was fabricated with less than 10 kg of magnets, which can accommodate samples up to a height of 70 mm. The 2D MPI has a spatial resolution of 3.5 mm upon mechanically moving the FFL.

## 4. Conclusions

We introduced a novel FFL generator comprising pairs of Halbach arrays and bar magnets. The manufactured FFL generator shows a high gradient of 4.7 T/m at a 30 mm diameter cylindrical FOV with a 70 mm open bore and a total magnet weight of under 10 kg. In the MPI implementation of this FFL generator, the 2D MPI image obtained showed a spatial resolution of less than 3.5 mm. Goodwill et al. developed an MPI scanner with a field gradient of 2.35 T/m and a magnet free bore size of 10 cm. The MPI system has a resolution of 3.5 × 8.0 mm and an FOV of 2.5 cm × 5.0 cm [18]. Irfan et al. presented a study on a selection field generation system that can be used for MPI with a selection field size of 4.3 T/m along three axes using a 10 cm-sized permanent magnet [21]. Although some studies have proposed solutions to generate a selection field of 7.0 T/m or more, it is difficult to find examples of actual application to MPI.

This result shows that the combination of the Halbach array and bar magnet pairs in the FFL generator is beneficial for building a compact and high-performance MPI system.

Mechanically driven MPI with the proposed FFL generator has a few advantages. Since the large amplitude and high-frequency fields required to generate selection fields and to shift the positions of the magnetic fields are not used in this MPI system, it is free from the health concerns that could be caused by high-energy alternating magnetic fields. It also has the advantage of being relatively safe and inexpensive to manufacture because it does not require a cooling system that is expensive to manufacture and maintain, and it uses small magnets.

We anticipate this approach to be applied to building MPI systems for diverse purposes, for example, not only to achieve a high-resolution MPI at a smaller FOV, but also to build medical MPI systems with larger FOVs, such as for human brain scanning, with an appropriate weight.

## Figures and Tables

**Figure 1 sensors-24-00933-f001:**
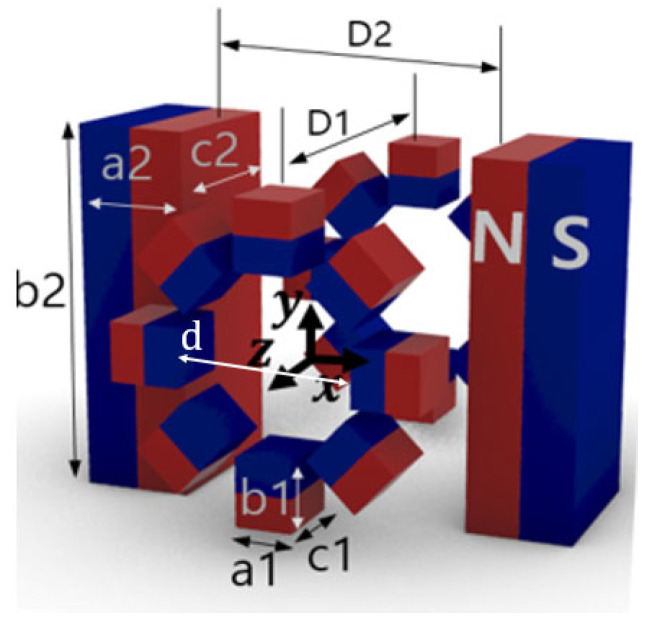
Configuration of the proposed FFL generator (d: inner diameter of Halbach ring, D1: distance between Halbach ring pair, D2: distance between bar-shaped magnet pair, [a1, b1, c1]: dimensions of small magnets of Halbach ring, [a2, b2, c2]: dimensions of bar-shaped magnets).

**Figure 2 sensors-24-00933-f002:**
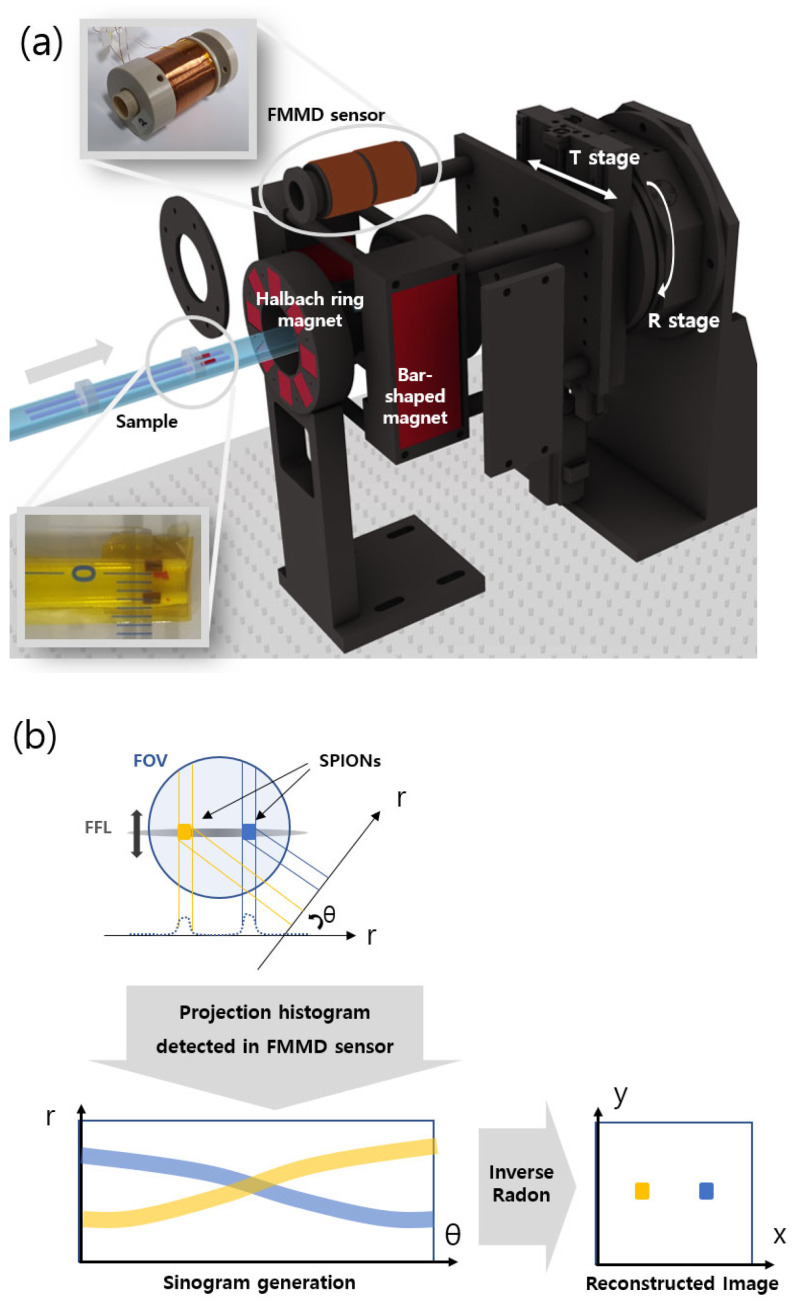
MPI system with the proposed FFL generator. (**a**) A 3D rendering of the setup, (**b**) schematic illustration of the measurement mode of rotating the FFL by an angle θ, yielding a sinogram from which the image is reconstructed.

**Figure 3 sensors-24-00933-f003:**
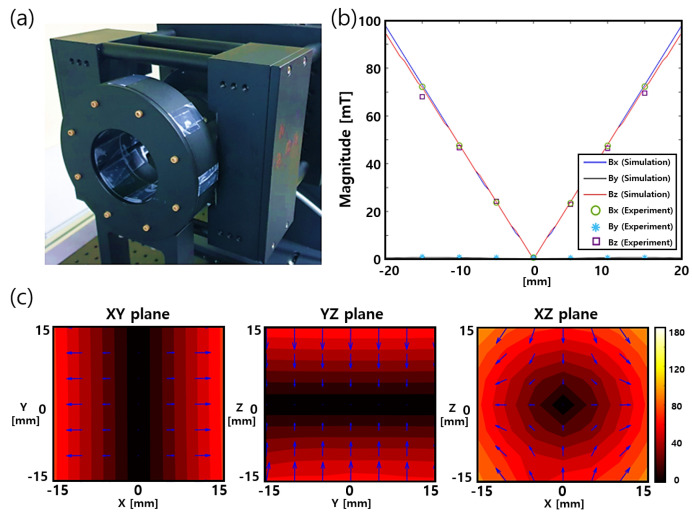
Experimental results of the FFL generator: (**a**) photograph of the crafted FFL generator with aluminum casing, (**b**) measured gradient strength compared with simulation results, (**c**) measured 2D contour graph of the magnetic field of the FFL generator.

**Figure 4 sensors-24-00933-f004:**
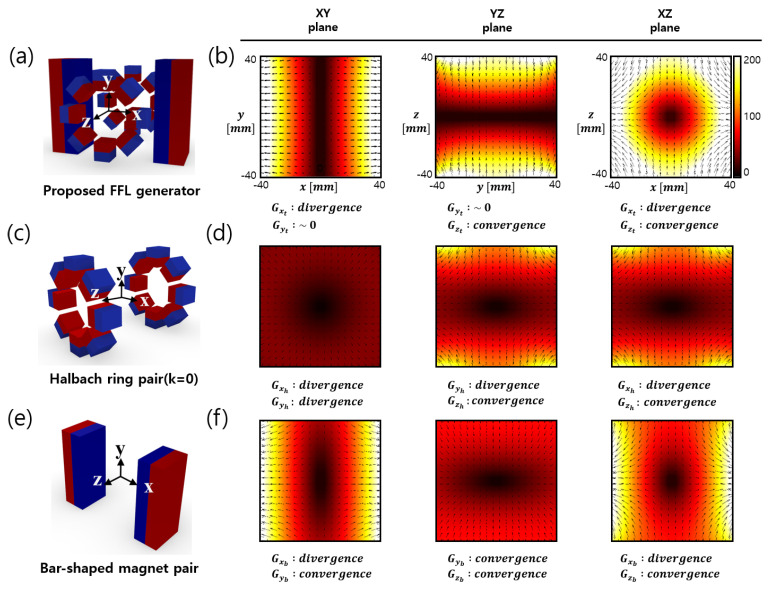
Proposed FFL generator and FFL characteristics. (**a**) Schematics of the FFL generator, (**b**) FFL 3D contour graph of the FFL generator, (**c**) magnet arrangement of a pair of Halbach rings, and (**d**) FFL 3D contour graph of its field, (**e**) bar magnet pair, and (**f**) FFL 3D contour graph of its field.

**Figure 5 sensors-24-00933-f005:**
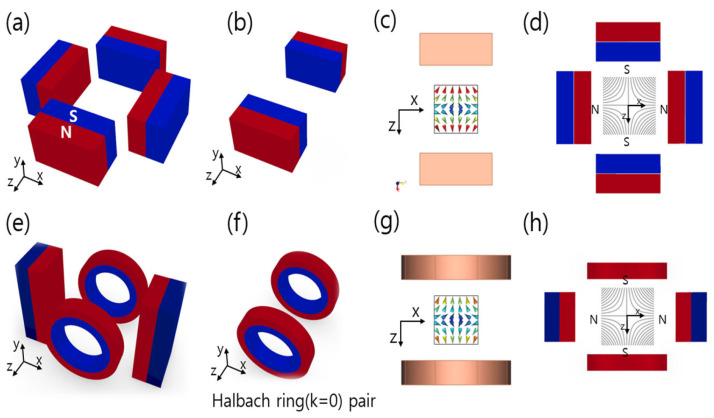
Comparison of quadrupole magnets and the proposed FFL generator: (**a**) quadrupole magnets, (**b**) square magnet pair, (**c**) magnetic field of square magnet pair, (**d**) magnetic field of quadrupole magnet, (**e**) proposed FFL generator, (**f**) Halbach ring pair, (**g**) magnetic field of the pair of Halbach rings, (**h**) magnetic field from the proposed FFL generator.

**Figure 6 sensors-24-00933-f006:**
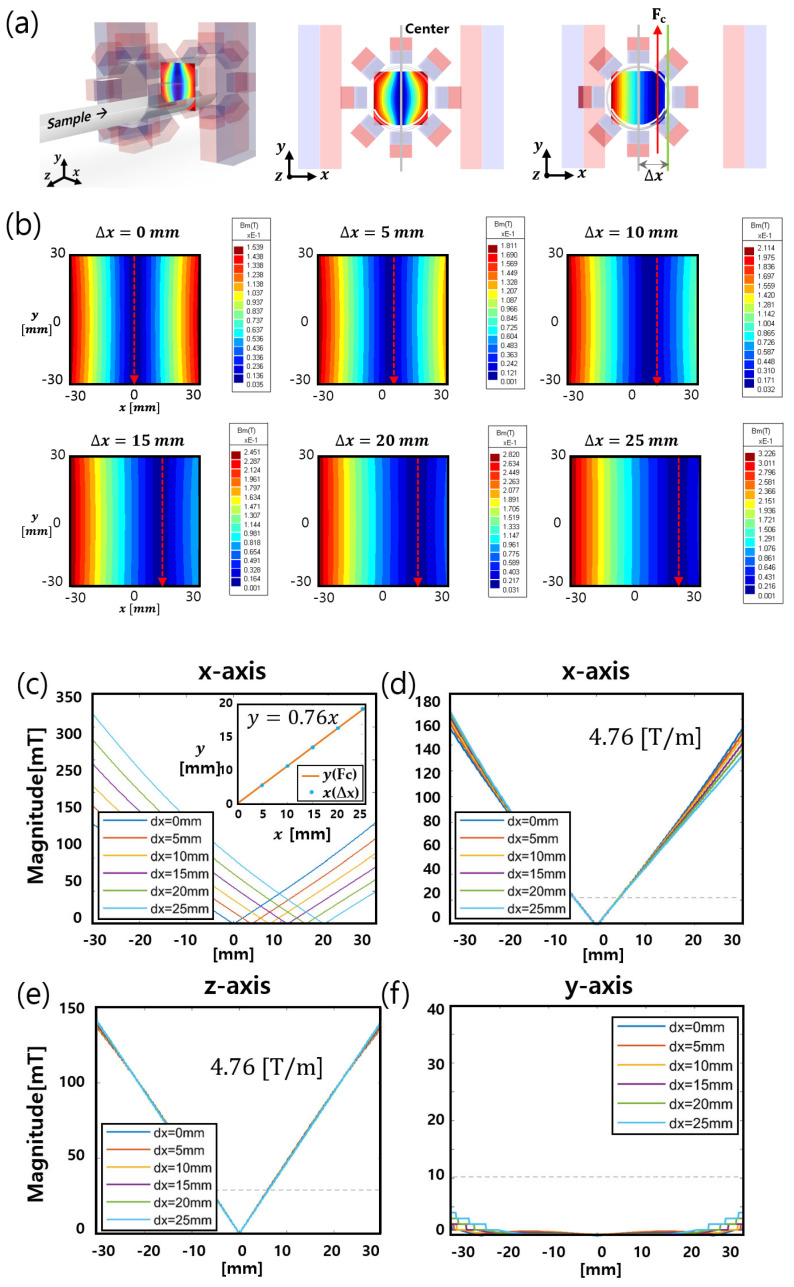
Linear movement of the FFL and gradient strength: (**a**) translation movement of the bar magnet pair, (**b**) contour map along the x-y plane, magnetic strength along the (**c**,**d**) *x* axis, (**e**) *z* axis, and (**f**) *y* axis while the bar magnet pair is displaced (∆x) from 0 mm to 25 mm.

**Figure 7 sensors-24-00933-f007:**
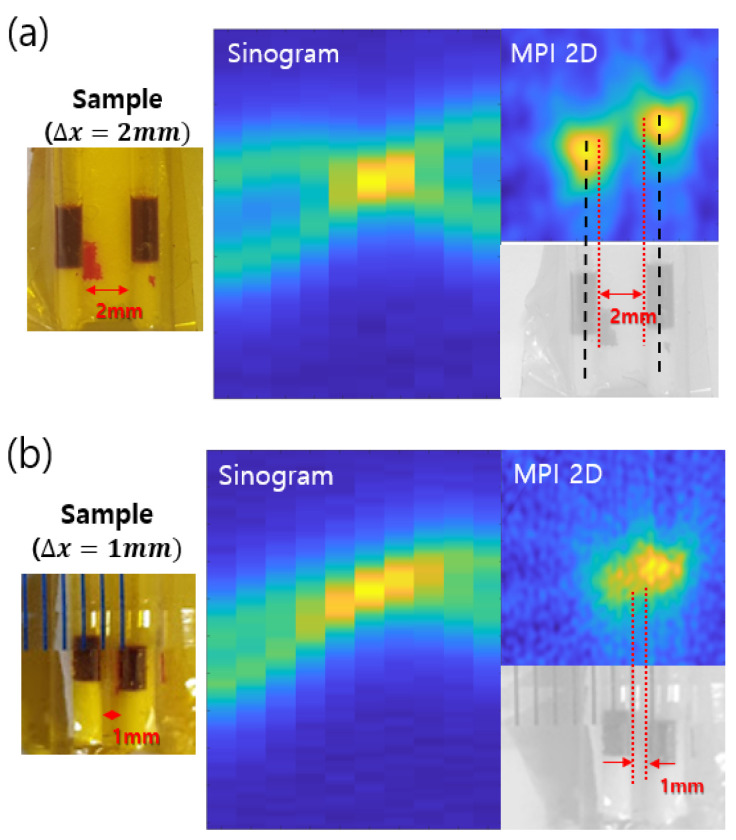
MPI results with sample distance (∆x) of (**a**) 2 mm and (**b**) 1 mm.

**Table 1 sensors-24-00933-t001:** Dimension of the FFL device with 70 mm open bore (d: inner diameter of Halbach ring, D1: distance between Halbach ring pair, D2: distance between bar-shaped magnet pair, n1: number of small magnets of Halbach ring, [a1, b1, c1]: dimensions of small magnets of Halbach ring, [a2, b2, c2]: dimensions of bar-shaped magnets).

Aperture Size [mm]	Geometry (d, D1, D2, n1, [a1, b1, c1], [a2, b2, c2]) [mm]	Magnetic Field Gradient (Simulation/Measured)	Magnet Weight (Total/Bar Magnets)	FOV
70	(74, 112, 70, 8, [30, 25, 30] [160, 70, 50])	4.76 T/m/4.70 T/m	9.55 kg/7.59 kg	30 × 30 × 100 mm^3^

## Data Availability

Data are contained within the article.

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
