# Peer review of "A Novel Field-Free Line Generator for Mechanically Scanned Magnetic Particle Imaging"

_sensors, 2024, doi:10.3390/s24030933_

Round 1

Reviewer 1 Report

Comments and Suggestions for Authors

In this work, the authors proposed an efficient field-free line (FFL) generator for mechanically driven FFL for magnetic particle imaging applications. The experimentally obtained results show that the proposed configuration for FFL have some advantages in terms of mass of used PM and manufacturing cost. The principles of the generation of the FFL and how to move (shift and rotate) it are clearly described and the manuscript, overall, is well written.

The followings are the questions and comments.

Questions and comments

1. (p. 3, line 88) The product name and the company name are missing.

2. (p. 9, line 263) I guess some words are missing.

3. (Fig. 3(b), Fig. 6) Please use high resolution images.

4. What is the speed of translational movement and rotation of the FFL for MPI imaging experiment shown in Fig. 7?

Reviewer 2 Report

Comments and Suggestions for Authors

Dear Authors,

Thank for your interesting manuscript. The MPI is an frontier topic of the medical imaging techniques. Many scientific groups are working in this area and trying to improve its characteristics, including spatial and temporal resolution. I believe the manuscript can be suitable for publication in the Journal after revising some issues.

1. The Authors pay great attention to the high achievable spatial resolution of MPI. Nevertheless, the current values of this parameter has not been given in the Introduction. Some papers tell us about the spatial resolution of 1 mm or even less when using SPIONs, therefore I recommend to give more references in the Introduction to stress the novelty of the current study. 

2. The comparison of MPI with CT and MRI technique has to be enhanced and given in more detail. The Authors point, that "MRI entails a substantially long examination time" (Line 53), but no references are given to prove the statement.

3. The obtained field gradient (4.76 T/m) is high, but not the highest. Some studies report about even higher values (for example, see https://doi.org/10.1080/02656736.2020.1853252, https://doi.org/10.1039/C7NR05502A or the other papers). Therefore, the Authors are recommended to compare their own results with the previously obtained.

4. The same comparison is recommended for the achieved spatial resolution, since there are some studies reporting on the higher spatial resolution (see for example https://doi.org/10.1038/srep34180,   https://doi.org/10.3390/nano9101466 or the other papers).

5. The SPIONs size, composition and structure can affect the MPI parameters. The Authors use some commercial SPIONs (Line 100) and do not provide their characteristics. It is recommended to be added to the Materials and Methods section.

6. The 3-axis Hall-effect sensor should be specified (Line 87). In the current revision of the manuscript the phrase "3-axis Hall-effect sensor (name, company)" is only given. The detailed technique of the magnetic field gradient estimation has to be given in more detail. 

7. The text in some panels of Figures 3, 4 and 6 is too small and therefore unreadable. I recommend to make the text larger or to redraw the figures.

Round 2

Reviewer 2 Report

Comments and Suggestions for Authors

Dear Authors,

Thank for addressing my comments.

I believe the manuscript is now ready for publication.